# Healthcare providers' pain management practice and its associated factors in Ethiopia: A systematic review and meta- analysis

**Demewoz Kefale**[1]*, **Tigabu Munye Aytenew**[2], **Yohannes Tesfahun Kassie**[3], **Melese Kebede**[3], **Maru Mekie**[4], **Mahilet Wondim**[5], **Shegaw Zeleke**[2], **Solomon Demis**[6], **Astewle Andargie Baye**[2], **Keralem Anteneh Bishaw**[7], **Gedefaye Nibret**[8], **Yeshiambaw Eshetie**[2], **Zelalem Tilahun Muche**[9], **Habtamu Shimels**[6], **Muluken Chanie**[10], **Mastewal Endalew**[11], **Worku Necho Asferie**[6], **Amare kassaw**[1]

1 Department of Pediatrics and Child Health Nursing, College of Health Sciences, Debre Tabor, Ethiopia, 2 Department of Nursing, College of Health Science, Debre Tabor University, Debre Tabor, Ethiopia, 3 Department of Emergency and Critical Care Nursing, College of Health Sciences, Debre Tabor University, Debre Tabor, Ethiopia, 4 Department of Reproductive Health and Family Health, College of Health Sciences, Debre Tabor University, Debre Tabor, Ethiopia, 5 Department of Midwifery, South Gondar Zone Health Office, Debre Tabor, Ethiopia, 6 Department of Maternal and Neonatal Health Nursing, College of Health Science, Debre Tabor University, Debre Tabor, Ethiopia, 7 Department of Maternity and Reproductive Health Nursing, College of Health Sciences, Debremarkos University, Debre Markos, Ethiopia, 8 Department of Midwifery, College of Health Science, Debre Tabor University, Debre Tabor, Ethiopia, 9 Department of Medical Physiology, College of Health Sciences, Debre Tabor University, Debre Tabor, Ethiopia, 10 Department of Epidemiology and Biostatistics, Institute of Public Health, College of Medicine and Health Sciences, Gondar University, Gondar, Ethiopia, 11 Department of Environmental and Occupational Health and Safety, Institute of Public Health, College of Medicine and Health Sciences, University of Gondar, Gondar, Ethiopia

* demewozk@yahoo.com

## Abstract

### Introduction

Pain is defined as an unpleasant sensory and emotional experience associated with actual or potential tissue damage. Under -treatment of pain continues to be a major health care problem in Ethiopia. Although it has this problem, it receives limited research attention. This systematic review and meta-analysis will provide valuable insights of prevalence and its factors of healthcare providers' pain management practice in Ethiopia.

### Methods

This systematic review and meta-analysis followed Preferred Reporting Items for Systematic Reviews and Meta-Analyses guidelines. Universal online databases such as PubMed, Cochrane, Google, Google Scholar, SCOPUS, Web of Science and Global Health were used to search for articles. Microsoft Excel for data extraction and STATA17 for data analysis were used. DerSimonian and Laird random-effects model was used to pool the odds ratios across studies and compute the overall pooled prevalence and its predictors. Egger's test with funnel plot symmetry and Cochrane's Q test was used to assess publication bias and heterogeneity respectively.

**Funding:** The author(s) received no specific funding for this work.

**Competing interests:** The authors have declared that no competing interests exist.

## Results

The overall pooled prevalence of health care providers' pain management practice was 39.6% (95% CI: 34.8, 44.5); $I^2$ = 97.0%; P<0.001). Availability of pain management protocol (AOR = 5.1, 95%CI: 3.6, 6.7), Accessibility of analgesia (AOR = 4.5, 95%CI: 1.96, 7.0), higher educational level (AOR = 3.3, 95%CI: 2.5, 4.1), being female (AOR = 1.2, 95% CI: 1.6, 3.1), took training (AOR = 2.7, 95% CI: 1.8, 3.6), decreased work load (AOR = 4.9, 95% CI: -1.9, 11.7), increased work experience (AOR = 3.9, 95% CI: 2.9, 5.1), Being midwifery profession (AOR = 2.5,95% CI: 1.6, 3.4), having good attitude (AOR = 3.9,95%CI: 2.5, 5.4) and being knowledgeable (AOR = 4.2, 95%CI: 2.8, 5.6) of health care providers towards pain management practice were significantly associated in Ethiopia.

## Conclusion

The overall prevalence of pain management practice among healthcare providers in Ethiopia is low compared with a national target of pain free hospital initiatives in Ethiopia. It needs a call to build health care providers' ongoing education, training, professional development and manageable workload.

## Introduction

Even though pain defined in different ways, the most popular is an international association for the study of pain which defined as an unpleasant sensory and emotional experience associated with actual or potential tissue damage [1]. Pain is a concealed sensation caused by high-intensity stimuli triggering pain receptors in the skin, muscle, bone, and other tissues which is now accepted as the fifth important cardinal sign [2],which should be evaluated and treated regularly [3]. If it is not treated timely and accurately, It touches almost every part of the survival of the patients as well as the family [4]. Pain can classified as acute or chronic. Acute pain is rapid but diminishes under treatment. Whereas chronic pain becomes progressively worst and reoccurs intermittently [5]. Clinical manifestation of pain is a highly subjective and deeply personal experience which has difficulty of concentrating, lack of energy, lost productivity, decreased quality of life and inability to complete daily tasks [6]. A thorough pain assessment, which, itself, may be the most useful step in planning to optimal treatment strategy for the patient [7]. The prevention and management of pain is an important aspect of health care [8]. Healthcare providers' play a key role in pain assessment and in counseling on the standards of pain management in different health care service units. They are the primary health care providers for the patients within different care settings [9]. In ill patients effective management of pain is vital to ensure the best possible clinical consequences and avert unnecessary stay in hospitals and hurt from pain [10].

Incidence of pain is in elevation in developing countries due to late diagnosis of disease and major barriers to opioid access. In case of life-threatening illness, pain and its treatment are forgotten, or under-appreciated by the health care providers [11].

Globally, the percentage of persons suffering pain in 2022 (34.1%) stayed elevated compared to the previous year in 2019 (33.3%) [12]. In Africa, patients suffer from postoperative pain accounts around 95.2% which is challenging to manage it [13]. Researches conducted in developing nations revealed that health care providers' pain management practice is inconsistent 24%, 52.7%, in Nepal respectively [14, 15]. A recent study conducted in Ghana and Kenya

showed that healthcare providers' pain management practice is 57.8% [16] and 9% [17]. And primary study conducted in Ethiopia showed with substantial differences from13% [18] to 66% [19]. This fluctuating finding revealed that how much pain is a neglected problem in the low-resource setting [20].

A Global survey which conducted to evaluate Pain management practices provided that religion, lack of education or training, non-adherence to guidelines, over regulation associated with prescribing and access to opioid analgesics, fear of addiction to opioids are key barriers among healthcare providers' [21]. Primary study conducted in Ethiopia showed that Pain management has been unaddressed due to factors like limited resources, inadequate training, cultural diversity, emotional feelings, physical characteristics and language barriers to receive basic pain care for patients [22]. Furthermore, Availability of pain management protocol, Accessibility of analgesia, educational level, sex, training work load, work experience, profession, attitude and knowledge of health care providers towards pain management practice were significantly associated in Ethiopia [23–26]. Overlooking patients' pain leads to many consequences and complications for both the patients and the healthcare organization [27]. The pain management practices in any healthcare system were affected by three major barriers which include organizational barriers, healthcare providers' barrier and patients' barriers [28].

Pain management tended to accentuate the habit of pharmacological agents, but intensive usage of analgesics can have negative repercussions as it can significantly affect some physiological functions, side effects, drug dependency and increasing health care prices and later on country's economy burden [29].

Although, world health organization, launched physical, psychological and pharmacological interventions for pain, healthcare providers' practicing level needs further attentions [30]. Even though, there are different attempts to increase pain management practice in Ethiopia like, Pain-free hospital implementation, interdisciplinary pain medicine and palliative care program and taking pain as a fifth vital sign [31–33], improvement in the assessment and treatment of pain is still in unacceptable change. Pain reliving intervention is an ignored and neglected especially in low- and middle-income countries including Ethiopia [34, 35].

Therefore, determining national prevalence and predictor factors of pain management practice among healthcare providers' in Ethiopia has pivotal role towards achieving Sustainable developmental goals. But doing research, financial allocation, policy initiatives and public awareness is a big home work for developing countries including Ethiopia. Therefore, this systematic review and meta-analysis assessed the pooled prevalence and associated factors of healthcare providers' pain management practice. Furthermore, this study will offer relevant information for future researchers, policymakers, clinicians, and concerned stakeholders.

## Method

### Study design and search strategy

This systematic review and meta-analysis were conducted to estimate the pooled prevalence, and associated factors of pain management practice among health care providers in Ethiopia. The protocol was designed based on the Standard Preferred Reporting Items for Systematic Review and Meta-analysis (PRISMA) guidelines [36] was used to report the results of this systematic review and meta-analysis (**S1 File**). Prior to the completion of this review, the protocol was prepared and submitted to the International Prospective Register of Systematic Reviews database (PROSPERO, 2024: CRD42024512855).

## Databases and search strategy

We have extensively searched PubMed, Google Scholar, Web of Science, Google, Global health, scopus and Chochrane databases for all available primary studies reporting health care providers pain management practice and its predictors in Ethiopia using the following search terms and phrases(**S2 File**)"(((("Healthcare providers ") AND ("pain")) AND ("prevalence or incidence")) AND ("factors") AND Ethiopia, "Healthcare providers OR professionals AND pain management practice,-, Burden of healthcare providers pain management practice and Ethiopia, prevalence or incidence" pain management practice and Ethiopia, Healthcare providers pain management practice in Ethiopia, Health care providers AND pain management practice And Ethiopia, "Health care providers", "obstetrics care providers" OR "health professionals "OR Nurses OR midwives OR Anesthesia OR General practitioners OR specialists OR Integrated emergency surgical officers AND Ethiopia)″. The search string was developed using ″AND″ and ″OR″ Boolean Operators. Moreover, a manual search of the reference lists of included studies was also performed. The searched studies were published between 2014 and 2023 in Ethiopia and published in English.

## Eligibility criteria

This systematic review and meta-analysis used condition, context, and population (CoCoPop) framework due to that all observational (cross-sectional) studies were included which conducted between 2014 and 2023 in Ethiopia among health care providers' pain management practice and its associated factors. Studies which reported prevalence of healthcare providers pain management practice, and/or at least one factors influencing healthcare providers pain management practice) which is conducted in Ethiopia) among healthcare providers that written in English were eligible. However, citations without abstracts, full texts, anonymous reports, editorials, systematic reviews and meta-analyses and qualitative studies were excluded from the review.

## Study selection

All the retrieved studies were exported to EndNote version 7 reference manager and the duplicated studies were removed. Initially, three independent reviewers (DK, TM and SZ) screened the titles and abstracts, followed by the full text reviews to determine the eligibility of each study. The disagreement between the two reviews was solved through discussion.

## Data extraction

Two independent reviewers (DK and SD) have extracted the data using a structured Microsoft excel data extraction form. Whenever variations were observed in the extracted data, the phase was repeated. When the discrepancies between the data extractors were continued, the third reviewer (WN) was involved. The name of the first author and year of publication, region, study area, study design, sample size, response rate and effect size of the included primary studies were extracted.

## Primary outcome measure of interest

The primary outcome of interest was prevalence of pain management practice and its associated factors among health care providers in Ethiopia. The odds ratio (OR) and standard error (SE) were calculated as effect measures for the study of prevalence and its determinant factors, respectively.

## Quality assessment and appraisal of the included studies

The quality assessment of included studies is indeed crucial for this systematic review and meta-analysis. The Quality of each study was evaluated using the Joanna Briggs Institute (JBI) quality appraisal criteria using the prevalence; cross-sectional Joanna Briggs Institute (JBI) quality appraisal checklist. Two independent reviewers (DK and TM) appraised the quality of the included studies, and scored for the validity of results. The quality of each study was evaluated using the Joanna Briggs Institute (JBI) quality appraisal criteria [37]. All studies were appraised using JBI checklist for cross-sectional studies [18, 22–26, 38–61]. Accordingly, among all thirty eight cross-sectional studies, thirty studies scored seven of nine questions, 77.8% (low risk), five studies scored six of nine questions, 66.7%(low risk), and the residual three studies likewise scored five of nine questions, 55.6% (low risk)(**Table 1**). Studies were considered to be of low risk when they scored 50% or advanced on the quality assessment indicators. After conducting an exhaustive quality appraisal, we resolute that the primary studies involved in our analysis showed a high level of reliability in their procedural quality scores. The cross-sectional studies scored between 5 and 7 out of a total of 9 points had high quality.

JBI's critical appraisal Checklist for the included studies for the systematic review and meta-analysis of healthcare providers' pain management practice and its associated factors in Ethiopia (**Table 1**)

## Data analysis

STATA version 17 statistical software was used to analyze all the statistical analyses. A weighted inverse-variance random-effects model [62]was used to compute the overall pooled prevalence of pain management practice among health care providers and determine the impact of its predictors. The presence of publication bias was checked by observing the symmetry of the funnel plot, and Egger's test with a p-value of <0.05 was also employed to determine a significant publication bias [63]. The percentage of total variation across studies due to heterogeneity was assessed using $I^2$ statistics [64]. The values of $I^2$ 25, 50 and 75% represented low, moderate and high heterogeneity respectively [64]. A p-value of $I^2$ statistic<0.05 was used to declare a significant heterogeneity [65, 66]. To identify the influence of a single study on the overall meta-analysis, one-at-a-time method sensitivity analysis and Method of pain management, sample size, types of pain, years of publication and profession types sub-group analysis was performed. Subgroup analysis identify whether there is variability (heterogeneity) in the treatment effects across different subgroups of the population studied. A forest plot was used to estimate the effect of independent factors on the outcome variable and a measure of association at 95%CI was reported. The Odds Ratio (OR) was the reported measure of association in the eligible primary studies. To estimate the pooled OR effect, either a fixed-effects or a random-effects model is used. A fixed-effects model is used if all the included studies used comparable methodology and were from identical populations, whereas a random-effects model is used when the included studies used different methodologies and sampled from different populations. In our review, the included primary studies used different methodologies and drawn from several independent populations. Thus, a random-effects model was used for this study.

# Results

## Search results

The search strategy retrieved a total of 2208 studies from PubMed (n = 1000), Google Scholar (n = 500), Cochrane (n = 390), Web of Science (n = 36), Google (n = 17), Global Health (n = 7), Scopus (n = 258) studies. After carefully removing irrelevant studies based on their

**Table 1. General characteristics of the included studies for the systematic review and meta-analysis of healthcare providers' pain management practice and its associated factors in Ethiopia.**

| Primary studies | JBI's critical appraisal questions | | | | | | | | | Overall quality score (100%) | Included |
|---|---|---|---|---|---|---|---|---|---|---|---|
| | Q1 | Q2 | Q3 | Q4 | Q5 | Q6 | Q7 | Q8 | Q9 | | |
| Getu.AA, et al., 2020 | Y | Y | Y | N | Y | Y | Y | N | Y | 77.8 | √ |
| Tsegaye.D, et al., 2023 | N | Y | Y | Y | Y | Y | N | Y | N | 66.7 | √ |
| Bishaw.AK, et al., 2020 | Y | N | Y | Y | Y | Y | N | Y | Y | 77.8 | √ |
| Wassihun.B, et al., 2022 | Y | N | Y | Y | Y | Y | N | Y | Y | 77.8 | √ |
| Eyeberu.A, et al., 2022 | N | N | Y | Y | Y | Y | N | Y | N | 55.6 | √ |
| Dile.M, et al., 2016 | Y | Y | Y | Y | N | Y | N | Y | Y | 77.8 | √ |
| Ganta. M, et al., 2021 | N | Y | Y | Y | Y | Y | Y | Y | N | 77.8 | √ |
| Belay.ZM and Yirdaw.TL, 2022 | Y | Y | Y | N | Y | Y | Y | N | Y | 77.8 | √ |
| Sahile.E, et al., 2017 | N | Y | Y | Y | Y | Y | N | Y | Y | 77.8 | √ |
| Shiferaw.A et al., 2022 | N | Y | Y | Y | Y | Y | N | Y | Y | 77.8 | √ |
| Zeleke. S, et al., 2021 | Y | Y | Y | Y | N | Y | N | Y | Y | 77.8 | √ |
| Abdella.J, et al., 2021 | Y | Y | Y | Y | N | N | Y | Y | Y | 77.8 | √ |
| Kassa.MG et al,— 2014 | Y | Y | Y | Y | Y | Y | N | N | Y | 77.8 | √ |
| Tadesse.N et al., 2022 | Y | Y | Y | Y | Y | Y | N | N | Y | 77.8 | √ |
| Zeleke GY et al., 2023 | Y | N | Y | Y | Y | Y | N | Y | Y | 77.8 | √ |
| Wondimagegn GZ et al., 2021 | Y | Y | N | Y | Y | Y | N | Y | Y | 77.8 | √ |
| Wari.G et al., 2021 | Y | Y | Y | Y | N | Y | N | Y | Y | 77.8 | √ |
| Wurjine HT et al., 2018 | N | N | Y | Y | Y | Y | Y | Y | Y | 77.8 | √ |
| Mekdes.G, et al., 2017 | Y | Y | Y | Y | Y | Y | Y | N | N | 77.8 | √ |
| Fekede.L et al., 2023 | Y | Y | Y | N | N | N | N | Y | Y | 55.6 | √ |
| Berihun.B, et al., 2024 | Y | Y | Y | Y | Y | N | N | Y | Y | 77.8 | √ |
| Negash.TT, et al., 2022 | Y | Y | Y | Y | Y | N | N | Y | Y | 77.8 | √ |
| Eyeberu et al., 2023 | N | N | Y | Y | Y | Y | N | Y | N | 55.6 | √ |
| Wakgari.N, et al., 2020 | Y | N | Y | Y | N | Y | Y | Y | Y | 77.8 | √ |
| Miftah.R, et al., 2017 | Y | Y | Y | Y | N | Y | N | Y | Y | 77.8 | √ |
| Solomon.TE, et al., 2021 | Y | N | N | Y | Y | Y | Y | Y | Y | 77.8 | √ |
| Terfasa.AE, et al., 2022 | N | N | Y | Y | Y | Y | Y | Y | Y | 77.8 | √ |
| Roga et al., 2023 | Y | Y | Y | Y | Y | Y | N | N | N | 66.7 | √ |
| Tadesse. F et al., 2016 | N | N | Y | Y | Y | Y | N | Y | Y | 66.7 | √ |
| Dechasa.A et al., 2022 | Y | Y | Y | Y | Y | Y | Y | N | N | 77.8 | √ |
| Sefefe.MW, et al., 2021 | Y | Y | Y | Y | Y | N | N | Y | N | 66.7 | √ |
| BEYENE.B, 2019 | Y | Y | Y | Y | Y | Y | N | N | Y | 77.8 | √ |
| Mekonen.MW, et al., 2022 | N | Y | Y | Y | N | Y | N | Y | Y | 66.7 | √ |
| Negewo.NA, et al., 2020 | Y | Y | N | Y | Y | N | Y | Y | Y | 77.8 | √ |
| Foto.LL, et al., 2023 | Y | Y | Y | N | Y | N | Y | Y | Y | 77.8 | √ |
| Melesse.TG, et al., 2021 | Y | Y | Y | Y | N | Y | N | Y | Y | 77.8 | √ |
| Alelgn. Y, et al., 2022 | Y | Y | Y | Y | Y | N | N | Y | Y | 77.8 | √ |
| Emiru.A, 2015 | N | N | Y | Y | Y | Y | Y | Y | Y | 77.8 | √ |

Y: Yes, N: No, U: Unclear, Q: Question. The overall score is calculated by counting the number of Y in each row. Q1: Was the sample frame appropriate to address the target population? Q2: Were study participants sampled in an appropriate way? /Are the patients at a similar point in the course of their condition/illness? Q3: Was the sample size adequate? Q4: Were the study subjects and the setting described in detail?/Are confounding factors identified and strategies to deal with them stated? Q5: Was the data analysis conducted with sufficient coverage of the identified sample? /Are outcomes assessed using objective criteria? Q6: Were valid methods used for the identification of the condition? Q7: Was the condition measured in a standard, reliable way for all participants? Q8: Was there appropriate statistical analysis? /Were outcomes measured in a reliable way? Q9: Was the response rate adequate, and if not, was the low response rate managed appropriately?

titles and abstracts (n = 1620) and duplicated studies (n = 122), a total of 466 studies were selected for full-text review. Afterward, full-text reviews were conducted, resulting in the removal of 428 studies due to lack of complete texts, not written in English, conducted outside of Ethiopia, different target groups and the outcomes not well defined. Finally, 38 studies were found relevant to determine the prevalence of pain management practice and its predictors among health care providers. We traced the PRISMA flow chart [67] to show the selection process from initially identified records to finally included studies (**Fig 1**).

## Characteristics of the included studies

All thirty eight studies were conducted using cross-sectional study design [18, 19, 22–26, 38–61, 68–74]. Regarding geographical region, thirteen studies in Amhara [25, 38–40, 43, 45–48, 55–57, 75] Ten studies were conducted in Oromia [19, 23, 49, 54, 59, 61, 68, 69, 75], Seven studies in Southern nations nationalities [26, 41, 44, 58, 71, 72, 70], six studies in Addis Ababa [18, 50–53, 71], two studies in Tigray [22, 73],The total sample size of the included studies was

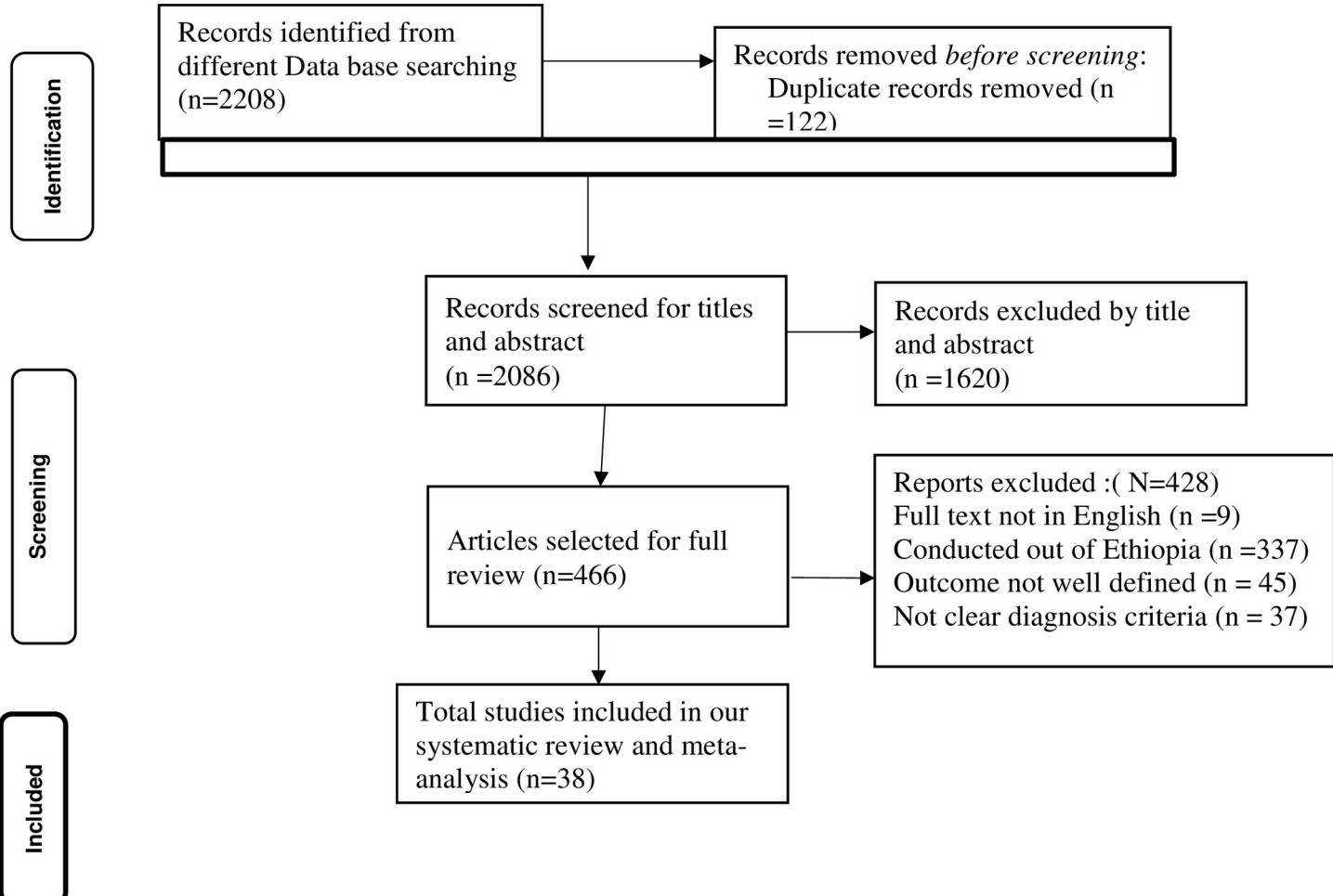

**Fig 1. PRISMA flow diagram of the included studies for the systematic review and meta-analysis of the healthcare providers' pain management practice and its associated factors in Ethiopia, 2024.**

11,844, where the smallest sample size was 82 in Addis Ababa [24] and the largest sample size was also 605 in Amhara region health research institute [38]. The prevalence of health care providers' pain management practice was obtained from thirty eight included primary studies [18, 19, 22–26, 38–61, 68–74].Which ranges from 13 to 66 [18, 19].While the data regarding the predictors of health care providers pain management practice were obtained from all included studies [18, 19, 22–26, 38–61, 68–74] (**Table 2**).

### Risk of bias assessment

The assessment tool [76] was used to assess the risk of bias. It consists of nine items that assess four areas of bias: internal validity and external validity. Items 1–4 evaluate selection bias, non-response bias and external validity. Items 5–9 assess measure bias, analysis-related bias, and internal validity. Accordingly, of the total of the thirty eight included studies, thirty studies scored seven of nine questions and the eight studies also scored five and six of nine questions. Studies were classified as ″low risk″ if five and above of nine questions received a ″Yes″, as ″moderate risk″ if 4.5 of nine questions received a ″Yes″ and as ″high risk″ if lower of 4.5 of nine questions received a ″Yes″. Therefore, all included studies [18, 19, 22–26, 38–61, 68–74] had low risk of bias (high quality).

### Meta-analysis

**Health care providers pain management practice.**   Subsequently, 38 eligible primary studies [18, 19, 22–26, 38–61, 68–74] were included in the final meta-analysis. In Ethiopia, the prevalence of health care providers pain management practice ranges from 13% in Addis Ababa [18] to 66% in Oromia [19] and the overall pooled prevalence of health care providers' pain management practice was 39.62% (95% CI:34.78, 44.47); $I^2 = 97.00\%$; P<0.001) (**Fig 2**).

**Publication bias among the studies.**   The symmetrical distribution of the included primary studies on the funnel plot suggests the absence of publication bias (**Fig 3**), and the p-value of Egger's regression test (P = 0.9032) also indicated the absence of publication bias.

**Analysis of heterogeneity.**   The percentage of $I^2$ statistics of the forest plot indicates a marked heterogeneity among the included studies ($I^2 = 97.00\%$, P<0.001) (**Fig 2**). Hence, sensitivity analysis and sub-group analysis were performed to minimize the heterogeneity.

*Sensitivity exploration.* To decide the effect of a particular study on the overall meta-analysis, we used the one-at-a-time method sensitivity analysis. Then forest plot revealed that the estimate from a single study is closer to the pooled estimate, which inferred the absence of a single study effect on the overall pooled estimate. Thus, it has been verified that a solitary study has no significant effect on the overall outcome of the meta-analysis (**Fig 4**).

### Sub-group analysis

**Sub-group Analysis by the methods of pain management.**   The pooled prevalence of health care providers' pharmacological pain management practice was 38.3% (95% CI: 31.8, 44.7); $I^2 = 97.50\%$; P<0.001) whereas the pooled prevalence of health care providers' non-pharmacological pain management practice was 41.2% (95% CI: 34.9, 47.4); $I^2 = 94.9\%$; P<0.001). This finding revealed that non-pharmacological pain management practice of health care providers nearly equal to pharmacological pain management practice of health care providers. So both methods are equally utilized to a significant extent by healthcare providers to use non-pharmacological and pharmacological methods for managing pain.

**Sub-group analysis by the type of pain.**   Labor and general types of pain in this sub-group analysis has no difference for health care providers pain management practice (AOR = 41.8, 95%CI: 35.3, 48.2, AOR = 38.0, 95%CI: 30.8, 45.2).

**Table 2. Characteristics of the included studies for the systematic review and meta-analysis of Healthcare providers' pain management practice and its associated factors in Ethiopia.**

| ID | Author [Year] | Study area | Years of publication | Sample size | Prevalence | Quality |
|---|---|---|---|---|---|---|
| 1. | Getu.AA, et al., 2020 | Amhara | 2020 | 605 | 46.8 | Low risk |
| 2. | Tsegaye.D, et al., 2023 | Amhara | 2023 | 326 | 48.1 | Low risk |
| 3. | Bishaw.AK, et al., 2020 | Amhara | 2020 | 309 | 30.40 | Low risk |
| 4. | Wassihun.B, et al., 2022 | SNNP | 2022 | 272 | 37.5 | Low risk |
| 5. | Eyeberu.A, et al., 2022 | Harari | 2022 | 464 | 59.3 | Low risk |
| 6. | Dile.M, et al., 2016 | Amhara | 2016 | 220 | 40.1 | Low risk |
| 7. | Ganta. M, et al., 2021 | SNNP | 2020 | 419 | 32.7 | Low risk |
| 8. | Belay.ZM and Yirdaw.TL, 2022 | Amhara | 2022 | 403 | 51.3 | Low risk |
| 9. | Sahile.E, et al., 2017 | Tigray | 2017 | 239 | 43.3 | Low risk |
| 10. | Shiferaw.A et al., 2022 | Amhara | 2022 | 323 | 41.3 | Low risk |
| 11. | Zeleke. S, et al., 2021 | Amhara | 2021 | 169 | 26 | Low risk |
| 12. | Abdella.J, et al., 2021 | Oromia | 2022 | 422 | 53.8 | Low risk |
| 13. | Kassa.MG et al, 2014 | Addis Ababa | 2014 | 82 | 34.1 | Low risk |
| 14. | Tadesse.N et al., 2022 | Addis Ababa | 2022 | 245 | 38.1 | Low risk |
| 15. | Zeleke. GY et al., 2023 | Addis Ababa | 2023 | 141 | 36 | Low risk |
| 16. | Wondimagegn. GZ et al., 2021 | Addis Ababa | 2021 | 208 | 56.5 | Low risk |
| 17. | Wari.G et al., 2021 | Addis Ababa | 2021 | 119 | 32.2 | Low risk |
| 18. | Wurjine .HT et al., 2018 | Oromia | 2018 | 148 | 52.1 | Low risk |
| 19. | Mekdes.G, et al., 2017 | Amhara | 2017 | 318 | 60 | Low risk |
| 20. | Fekede.L et al., 2023 | Amhara | 2022 | 437 | 25.8 | Low risk |
| 21. | Berihun.B, et al., 2024 | Amhara | 2022 | 421 | 53.6 | Low risk |
| 22. | Negash.TT, et al., 2022 | Amhara | 2022 | 118 | 24.58 | Low risk |
| 23. | Eyeberu.A, et al., 2023 | Harari | 2023 | 464 | 50.9 | Low risk |
| 24. | Wakgari.N, et al., 2020 | SNNP | 2020 | 420 | 13.8 | Low risk |
| 25. | Miftah.R, et al., 2017 | Tigray | 2017 | 261 | 55.8 | Low risk |
| 26. | Solomon.TE, et al., 2021 | Amhara | 2021 | 336 | 57.14 | Low risk |
| 27. | Terfasa.AE, et al., 2022 | Oromia | 2022 | 399 | 46 | Low risk |
| 28. | Roga.EY et al., 2023 | Oromia | 2023 | 268 | 37.3 | Low risk |
| 29. | Tadesse. F et al., 2016 | SNNP | 2016 | 184 | 24.4 | Low risk |
| 30. | Dechasa.A et al., 2022 | Oromia | 2022 | 384 | 66 | Low risk |
| 31. | Sefefe.MW, et al., 2021 | Amhara | 2022 | 346 | 22.5 | Low risk |
| 32. | BEYENE.B, 2019 | Harari | 2019 | 422 | 33.6 | Low risk |
| 33. | Mekonen.MW, et al., 2022 | Oromia | 2022 | 206 | 16.3 | Low risk |
| 34. | Negewo.NA, et al., 2020 | Oromia | 2020 | 203 | 23.5 | Low risk |
| 35. | Foto.LL, et al., 2023 | SNNP | 2023 | 421 | 37 | Low risk |
| 36. | Melesse.TG, et al., 2021 | SNNP | 2021 | 415 | 43.3 | Low risk |
| 37. | Alelgn. Y, et al., 2022 | SNNP | 2022 | 376 | 41.3 | Low risk |
| 38. | Emiru.A, 2015 | Addis Ababa | 2015 | 331 | 13 | Low risk |

Abbreviations: SNNP, Southern nations, nationality of people

**Sub-group analysis by the sample size difference.** The sample size sub-group analysis showed that pooled prevalence of health care providers' pain management practice with >350 sample sizes [43.7, 95%CI: 36.0, 51.3, $I^2$ = 97.7%, P<0.001] almost equal with <350 sample sizes [36.9, 95%CI: 30.8, 43.1, $I^2$ = 96.1%, P<0.001] (**Table 3**).

**Sub-group analysis by year of publication.** The studies conducted between 2020 and 2023 (AOR = 41.8, 95%CI: 36.4, 47.2, $I^2$ = 96.1. %, P<0.001) sub-group analysis revealed that,

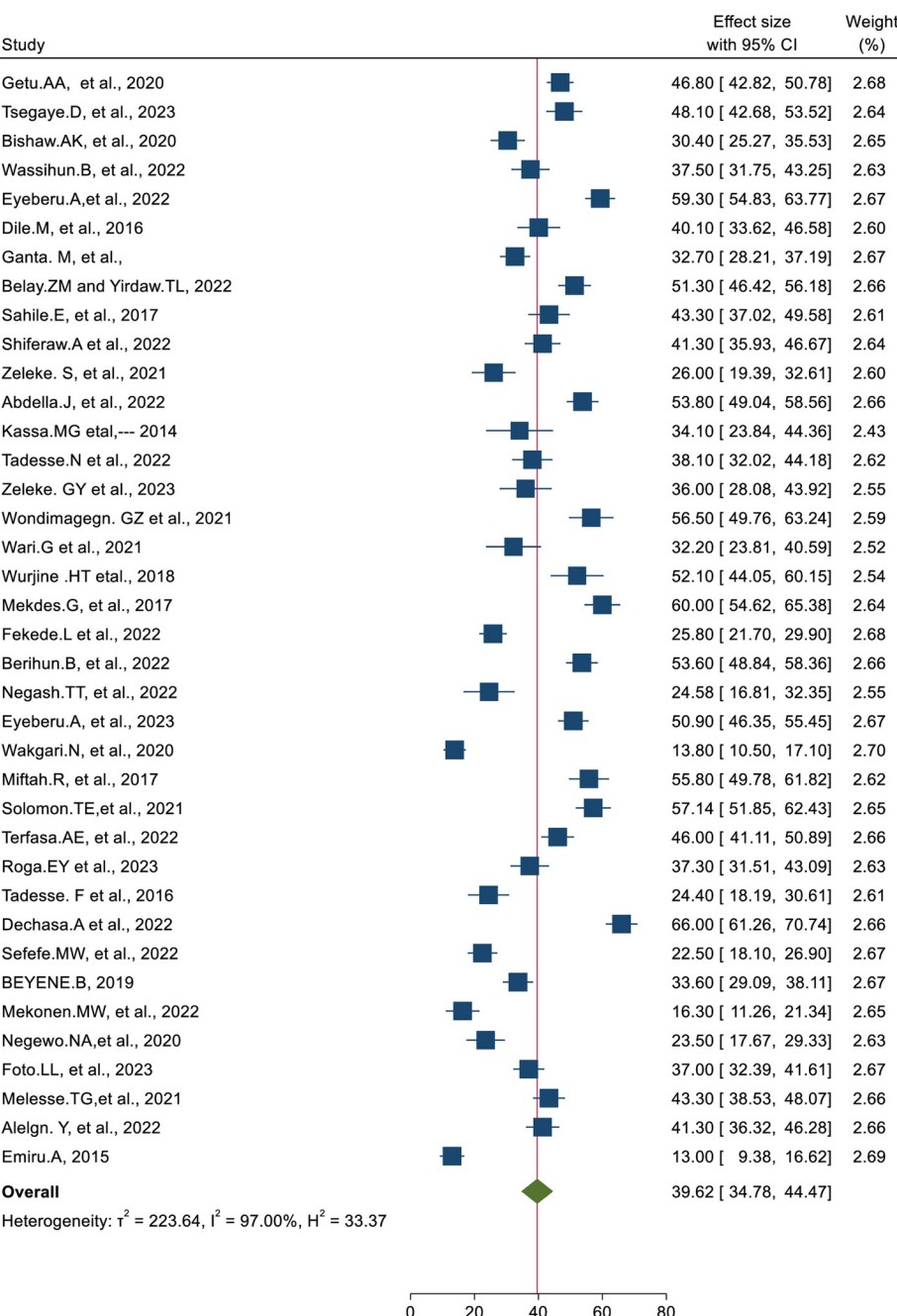

**Fig 2. Forest plot of the prevalence of the included studies for the systematic review and meta-analysis of health care providers' pain management practice in Ethiopia, 2024.**

no significant difference with study conducted before 2020 (AOR = 35.9, 95%CI: 27.5, 44.3, $I^2$ = 97.4%, P<0.001).

**Sub-group analysis by profession type.** The studies conducted on others (AOR = 32.1,95% CI;22.7,41.4, $I^2$ = 94.6%,P<0.00) revealed that low pain management practice compared to studies conducted among nurses only (AOR = 39.4,95% CI;30.8,47.9, $I^2$ =

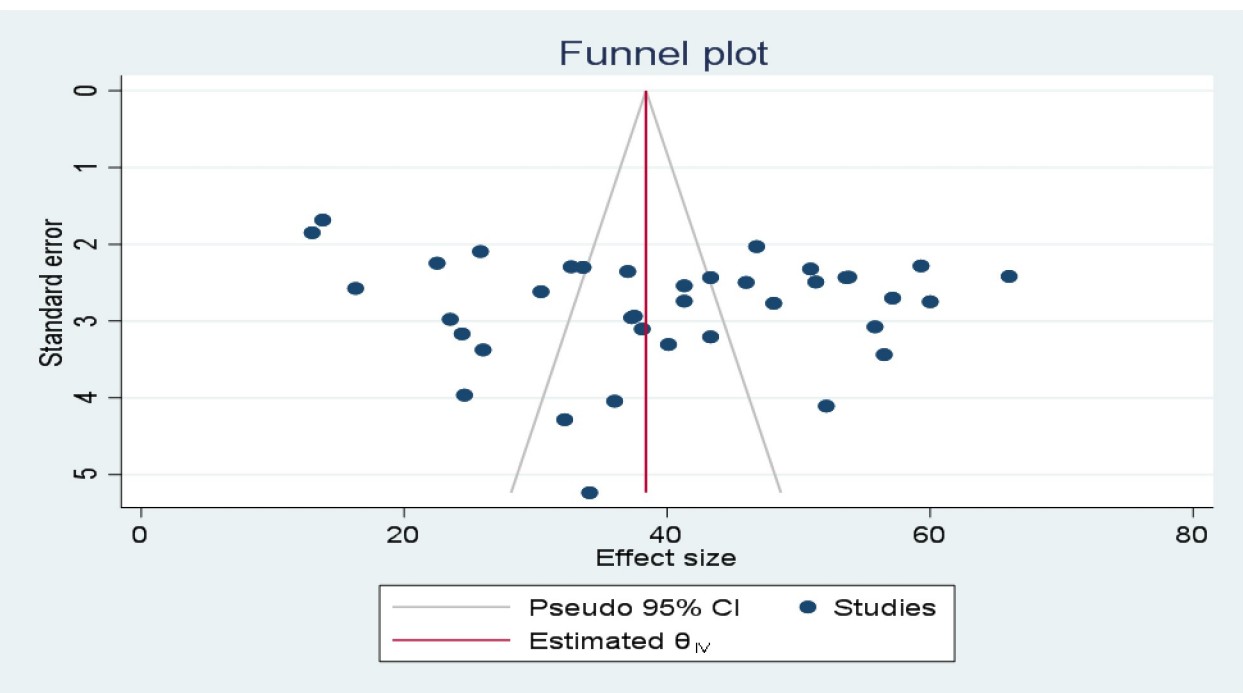

**Fig 3. Funnel plot to show the publication bias for included studies for the systematic review and meta-analysis of health care providers' pain management practice and its associated factors in Ethiopia, 2024.**

97.6%,P<0.00) and studies on nurses, midwifes and health officers (AOR = 42.9,95% CI;35.9,49.9, $I^2$ = 96.8%,P<0.00) respectively. This findings can conclude that Anesthesia, General Practionares and specialists have low pain management practice compared to nurses, midwifes and health officers.

## Predictors of health care providers pain management practice

In this review fifty studies [19, 23, 38, 39, 41, 44, 48, 56, 58, 59, 68, 71, 70, 73, 77], informed that health care providers' pain management practice was significantly associated with their attitude toward pain management practice. The pooled AOR of health care providers who has low attitude towards pain management practice was 3.93 (95%CI: 2.48, 5.39; $I^2$ = 98.92%; P<0.001) (**Table 4**).

Seven studies [19, 25, 38, 39, 47, 53, 68], showed a significant association between health care providers' pain management practice and availability of pain management protocol at their working setting. The pooled AOR of health care providers pain management practice for health care providers having pain management protocol was 5.13 (95%CI: 3.56, 6.71;$I^2$ = 99.59%; P<0.001) (**Table 4**).

Three studies [48, 72, 70] also reported a significant association between health care providers' pain management practice and accessibility of analgesia. The pooled AOR of health care providers pain management practice for health care providers who could accessed analgesia was 4.5 (95%CI: 1.96, 7.02;$I^2$ = 95.43%; P<0.001) (**Table 4**).

Eight studies [38, 40, 49, 68, 71–74], revealed that health care providers' pain management practice was positively associated with their educational level. The pooled AOR of health care providers pain management practice with their educational level was 3.3 (95%CI: 2.53, 4.13; $I^2$ = 92.00%; P<0.001) (**Table 4**).

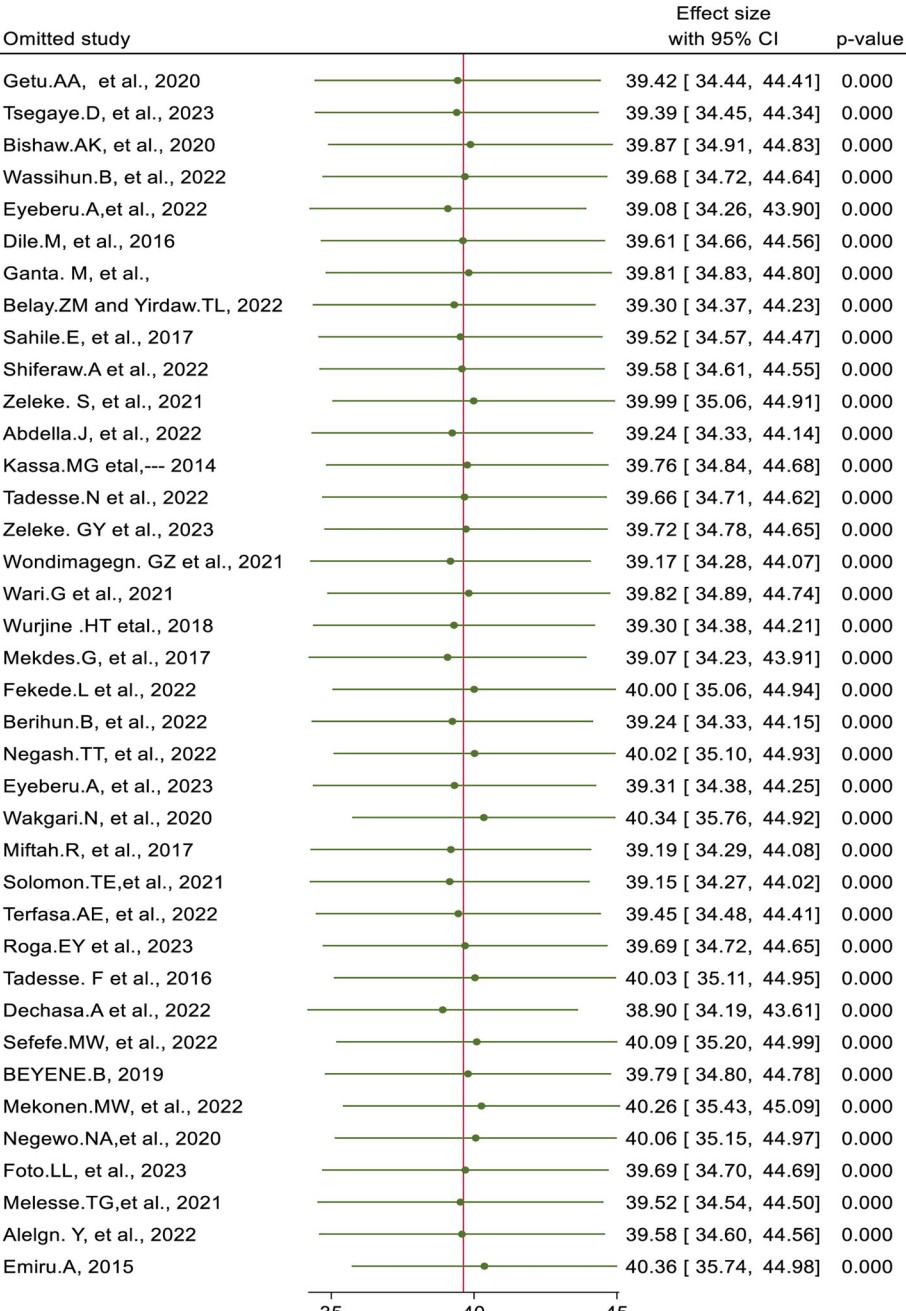

Fig 4. Forest plot to show after sensitivity exploration for included studies for the systematic review and meta-analysis of health care providers' pain management practice and its associated factors in Ethiopia, 2024.

To determine the association between health care providers' pain management practice and female sex, two studies were involved in the meta-analysis. Thus, the pooling of two studies [23, 72],showed that female health care providers were more likely to pain management practice than male health care providers [AOR = 1.24 [95% CI; 1.592, 3.062); $I^2$ = 85.96%, p < 0.001] (**Table 4**).

**Table 3. Subgroup analysis of the included studies for the systematic review and meta-analysis of healthcare providers' pain management practice and its associated factors in Ethiopia, 2024.**

| Variables | Outcome | Subgroup | No. of studies | Model | Prevalence | I² | P-value |
|---|---|---|---|---|---|---|---|
| Method of pain management | Pain management practice | Pharmacological | 26 | Random | 38.3% (CI:31.8, 44.7) | 97.5 | P < 0.001 |
| | | Non-pharmacological | 12 | Random | 41.2 (CI: 34.9, 47.4) | 94.9 | P < 0.001 |
| Sample size | Pain management practice | <350 | 23 | Random | 36.9 (CI:30.8,43.1) | 96.1 | P < 0.001 |
| | | >350 | 15 | Random | 43.7 (CI:36.1,51.3) | 97.7 | P < 0.001 |
| Types of pain | Pain management practice | Labor pain | 16 | Random | 41.8 (CI: 35.3, 48.2) | 96.5 | P < 0.001 |
| | | General pain | 22 | Random | 38.0 (CI: 30.8, 45.2) | 97.3 | P < 0.001 |
| Years of publication | Pain management practice | <2020 | 14 | Random | 35.9 (CI:27.5,44.3) | 97.4 | P < 0.001 |
| | | >2020 | 24 | Random | 41.8(CI:36.4,47.2) | 96.1 | P < 0.001 |
| Profession type | Pain management practice | Nurses only | 17 | Random | 39.4(CI:30.8,47.9) | 97.6 | P <0.001 |
| | | Nurses, midwifes and health officers | 15 | Random | 42.9(CI:35.9,49.9) | 96.8 | P < 0.001 |
| | | Others | 6 | Random | 32.1(CI:22.7,41.4) | 94.6 | P < 0.001 |

***Others** (nurses, midwifes, health officers, Anesthesia, General Practionares, specialists)

Fourteen studies were included in the meta-analysis to show the association between health care providers' pain management Knowledge and practice [19, 38, 44, 46–48, 52, 53, 56, 68, 69, 71, 72, 77]. As a result, the random effects model in the meta-analysis indicated that knowledgeable health care providers on pain management were more likely to practicing pain management than their counterparts (AOR = 4.2 (95% CI; 2.8, 5.6), I² = 98.62%, p < 0.001) (**Table 4**).

To compute the association between health care providers' pain management practice and the professional difference, four studies were selected for meta-analysis [40, 41, 59, 70]. The collective results of the study showed that midwifery professionals' were the most statistically associated with pain management practice than other professionals ([AOR = 2.5 (95% CI; 1.63, 3.44), I² = 89.73%, p <0.001]) (**Table 4**).

A pooled meta-analysis of eleven studies by means of a random effects model [19, 23, 48, 50, 52, 53, 59, 69, 72, 77, 78], showed that health care providers who had took training on pain management were more likely to performing appropriate means of pain management than who didn't take it([AOR = 2.7 (95% CI; 1.8, 3.6), I² = 93.67%, p < 0.001]) (**Table 4**).

**Table 4. Associated factors of the included studies for the systematic review and meta-analysis of healthcare providers' pain management practice in Ethiopia, 2024.**

| Variables | Outcome | No. of studies | Pooled Prevalence | Confidence interval | I² | P-value |
|---|---|---|---|---|---|---|
| HCPs' attitude | Factor | 15 | 3.93 | (95%CI:2.48, 5.39) | 98.9 | P<0.001) |
| Availability of PM protocol at their working setting | Factor | 7 | 5.13 | (95%CI: 3.56, 6.7) | 99.6 | P<0.001 |
| Accessibility of analgesia | Factor | 3 | 4.5 | (95%CI: 1.96, 7.0) | 95.4 | P<0.001 |
| HCPs' Educational level. | Factor | 8 | 3.3 | (95%CI: 2.53, 4.1) | 92.0 | P<0.001 |
| HCPs' sex | Factor | 2 | 1.24 | (95% CI; 1.59, 3.1) | 86.0 | P<0.001 |
| HCPs' Knowledge | Factor | 14 | 4.2 | (95% CI; 2.8, 5.6) | 98.6 | p < 0.001 |
| HCPs' Profession type | Factor | 4 | 2.5 | (95% CI; 1.63, 3.4) | 89.7 | p < 0.001 |
| HCPs' took training | Factor | 11 | 2.7 | (95% CI; 1.8, 3.6) | 93.7 | p < 0.001 |
| Work experience | Factor | 8 | 4.0 | (95% CI; 2.85, 5.1) | 94.2 | p < 0.001 |
| HCPs' work load | Factor | 3 | 4.9 | (95%CI;1.96, 11.7) | 99.6 | p <0.001 |

**PM**; Pain management, **HCPs'**: Healthcare providers'

Eight studies [26, 41, 49, 50, 56, 68, 70, 77], revealed that health care providers who has more work experience were more likely to practicing pain management than their counterparts([AOR = 3.98 (95% CI; 2.85, 5.11), $I^2$ = 94.23%, p <0.001]) (**Table 4**).This might be due to that experience by itself is the best teacher to know and to do pain management means.

Lastly a pooled meta-analysis of three studies by means of a random effects model [44, 48, 68] exhibited that heavy loaded health care providers has an obstacle to pain management practice of health care providers ([AOR = 4.87 (95% CI; 1.96, 11.70), $I^2$ = 99.56%, p < 0.001]) (**Table 4**).

## Discussion

According to this systematic review and meta-analysis, below half of all healthcare providers practicing appropriate pain management who needs much attention in Ethiopia. To the best of our knowledge, this is the first comprehensive national synthesis that gathered from different research with various forms of pain management methods have been broken down. Thus, this review intended to decide the overall pooled prevalence of health care providers' pain management practice and its predictors in Ethiopia. In this study, the overall pooled prevalence of health care providers pain management practice was 39.62% (95% CI: 34.78, 44.47); $I^2$ = 97.00%; P<0.001, which was higher than the study findings conducted in Kenya and Ethiopia (25.8%) [17, 78]. But the finding was lower than the study findings conducted in Spain (52.9%) [79],Nepal(67.5%) [80], Vietnam(72.2%) [81], India [82] and a study conducted in Ethiopia (45.73%,53.0%) [34, 83]. This discrepancy could be due to differences in study settings, methodologies, health care delivery systems across settings and the existence of socio-cultural variations.

In this review health care providers' level of knowledge has a significant associated factor for pain management implementation. This finding is consistent with a study conducted in USA [84].The possible explanation might be due to that practice is positively related to knowledge and it can affect health care providers' practicing level directly [85].

The results of this review and meta-analysis indicate significant associations between pain management practice among healthcare providers and different factors, including the availability of pain management protocols and analgesia [86], educational level, sex, training history, workload, work experience, profession type, attitude and knowledge towards pain management which is consistent with prior studies conducted in Brazil, USA, Saudi Arabia, and Ethiopia [87–91]. As previous research has concluded the practice of having health care providers' pain management strategies for the patients' can be categorized into three different models, the biomedical model, the non-pharmacological interventions model, and the alternative medicine model, including acupuncture [92–94], The identified factors play a pivotal role in advancing the level of pain management practices among healthcare providers. Training improves job satisfaction and morale, enhancing productivity and ultimately contributing to the advancement of healthcare providers' pain management practices.

Study findings need prompt action from health care leaders to educate and change the attitude of health care providers on pain management, which is supported by a study conducted [90, 95, 96].The possible reason might be due to that a positive healthcare providers attitude towards pain management is the manner in which they can conduct in a professional setting which contribute to the culture of your work environment, and how to perform daily tasks and responsibilities.

A cross-sectional study conducted in Nigeria reveled that healthcare providers' educational level was significantly associated with pain management practice [97].Due to that advanced education enhances their knowledge, skills, attitudes, and adherence to best practice. A Global

Survey which has been done on health care providers showed that religion, lack of appropriate education and training, non-adherence to guidelines, lack of access to opioid analgesics were significantly associated with healthcare providers pain management practice [21].

This study contributes to the growing body of evidence on healthcare providers' pain prevention and intervention by revealing its prevalence and factors. And it underlines the extensive impact of pain on patients' treatment outcomes and recovery rate which highlights the significance of addressing this intricate issue through multifaceted approaches. Furthermore, it elucidates the role of healthcare providers' knowledge, attitude, work load, work experience, profession type, sex and being trained and availability of analgesics factors in shaping the level of healthcare providers practicing pain intervention.

## Limitation of the study

The included studies' sample sizes varied greatly, which can have an impact on the pooled estimates' precision and statistical power. Although we used statistical tests and funnel plots to evaluate publication bias, it's likely that unpublished studies had an impact on the final outcomes. While subgroup analyses and sensitivity testing can mitigate study heterogeneity, differences in study design, demographic characteristics, intervention protocols, and outcome measures can still affect the reliability and generalizability of the pooled results. We assessed each study's risk of bias because several of the studies in the review had varied degrees of bias, but it's crucial to recognize that not all biases could be completely accounted for in the meta-analysis.

## Conclusions

The present study provides insight for health care providers' practices in treating patients with pain. The results direct the required for a more wide-ranging treatment approach to health care pain management practice. The overall pooled prevalence of pain management practice among health care providers was considerably low compared with a national target of pain free hospital initiatives launched in October, 2018 in Ethiopia. There is an urgent requisite to build health care providers' capability to enable ongoing education, training, professional development and manageable workload for best relief of pain management practice among patients.

## Supporting information

**S1 File. PRISMA checklist.**
(DOCX)

**S2 File. Search strategy.**
(DOCX)

## Acknowledgments

We are grateful to Debre Tabor University for providing internet services, which are vital for accomplishment of this systematic review and meta-analysis. Again, we extend our thanks to Mr. Wubet alebachew for his support in reviewing the manuscript. Moreover, our gratitude extends to Mr. Biniyam Minuye for his support in the English edition.

## Author Contributions

**Conceptualization:** Demewoz Kefale, Tigabu Munye Aytenew, Solomon Demis, Astewle Andargie Baye, Yeshiambaw Eshetie, Zelalem Tilahun Muche, Muluken Chanie, Amare kassaw.

**Data curation:** Demewoz Kefale, Solomon Demis, Muluken Chanie, Amare kassaw.

**Formal analysis:** Demewoz Kefale, Yohannes Tesfahun Kassie, Melese Kebede, Astewle Andargie Baye, Habtamu Shimels, Worku Necho Asferie, Amare kassaw.

**Funding acquisition:** Habtamu Shimels, Worku Necho Asferie.

**Investigation:** Mastewal Endalew, Worku Necho Asferie.

**Methodology:** Mastewal Endalew, Worku Necho Asferie.

**Software:** Demewoz Kefale.

**Validation:** Mahilet Wondim, Keralem Anteneh Bishaw.

**Visualization:** Demewoz Kefale.

**Writing – original draft:** Demewoz Kefale, Maru Mekie, Shegaw Zeleke, Amare kassaw.

**Writing – review & editing:** Demewoz Kefale, Maru Mekie, Gedefaye Nibret, Amare kassaw.

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
