## [Decision Letter · Decision Letter 0]

17 Jun 2024

PONE-D-24-12823HEALTHCARE PROVIDERS’ PAIN MANAGEMENT PRACTICE AND ITS ASSOCIATED FACTORS IN ETHIOPIA. SYSTEMATIC REVIEW AND META- ANALYSISPLOS ONE

Dear Dr. Mekonnen,

Thank you for submitting your manuscript to PLOS ONE. After careful consideration, we feel that it has merit but does not fully meet PLOS ONE’s publication criteria as it currently stands. Therefore, we invite you to submit a revised version of the manuscript that addresses the points raised during the review process.

We look forward to receiving your revised manuscript.

Kind regards,

Dereje Zewdu Assefa, BSc, MSc

Academic Editor

PLOS ONE

Journal Requirements:

Additional Editor Comments:

"The manuscript needs thorough language editing and proofreading." Please remember to add the search strategy as a supplementary file and provide details of the electronic database used, along with the corresponding article numbers found in each database.

Reviewers' comments:

Reviewer's Responses to Questions

**Comments to the Author**

1. Is the manuscript technically sound, and do the data support the conclusions?

Reviewer #1: Yes

Reviewer #2: Partly

2. Has the statistical analysis been performed appropriately and rigorously? 

Reviewer #1: Yes

Reviewer #2: Yes

3. Have the authors made all data underlying the findings in their manuscript fully available?

Reviewer #1: Yes

Reviewer #2: Yes

4. Is the manuscript presented in an intelligible fashion and written in standard English?

Reviewer #1: No

Reviewer #2: No

5. Review Comments to the Author

**Reviewer #1:** Comments for the author

Dear PLoS ONE team of editorials

Thank you for the chance given to me to review a manuscript entitled “Healthcare providers’ pain management practice and its associated factors in Ethiopia. Systematic review and meta- analysis: manuscript number “PONE-d-24-12823”. Since pain management is a fundamental aspect of patient care and quality of life, the finding of the manuscript in general has great clinical significance by providing information on the overall pictures of pain management status in Ethiopia and on the areas of improvement to enhance pain management practice of healthcare providers. The study upholds rigorous technical standards across statistical analyses, and methodological procedures. Please find my general and specific comments below.

General comments

The manuscript has multiple language usage flaws including punctuations, wordings, spelling and mainly grammar errors. These problems are found throughout the manuscript. . Therefore, please make repeated proof-reading and thorough copyediting before considering the manuscript for publication. This would help increase the readability of the manuscript if published..

For instance

Abstract: Pain defined need to be corrected as pain is defined as

Under treatment need to be corrected as under-treatment

Avoid unnecessary spelling in the following sentence. Advancing a national agenda to reduce inconsistencies in pain Care for health policy, education and Practice is very imperative

Inconsistencies in language usage e.g health care professionals’’ versus health care providers’’????

Work load need to be written as Workload

Specific comments

1. Title

The title needs little modification as follows. Healthcare providers’ pain management practice and its associated factors in Ethiopia: A systematic review and met analysis

2. Key words

I recommended the following as a key words. Pain management practice, health care providers, Ethiopia, Systematic review and meta-analysis

3. Abstract

The paragraph stating about the factors associated with the outcome variable need to be rewrite to make it legible and easily comprehended

The following conclusion is not supported by your finding. “Advancing a national agenda to reduce inconsistencies in pain Care for health policy, education and Practice is very imperative

4. Introduction

The introduction to shall be concentrated on health care providers pain management practice (definition, the figures on magnitude and the inconsistencies in the magnitude, the consequences of poor pain management in terms of quality of life , treatment outcome, healthcare cost…)

It would be better to add more information significance of the study with regard to the policy makers ( the significance of providing overall figures of pain management practice) , healthcare providers ( training and capacity building activities ) and patients ( in terms of reducing healthcare costs, hospitalization, improving quality of life…)

5. Methods and materials

Word in their first appearance in the manuscript need to be stated in the full explanation terms. GPs, IESO ..

On the Operational definition, it is better to add about the tool, the assessment criteria ..that your primary studies were utilized.

6. Result and Discussion

The discussion need to specifically stated for each independent variables

Please add the further implication of the finding (both on the prevalence and each associated factors ) in the discussion section.

Please refine the following paragraph accordingly. “The results of this review and meta- analysis identified that availability of pain management protocol and analgesia[76], educational level, female sex, took training, work load, work experience, type of profession, attitude and knowledge of health care providers towards pain management practice were significantly associated with pain management practice among health care providers. This results consistent with previous studies conducted in Brazil, USA, Saudi Arabia and Ethiopia [77-81]. As previous research has concluded the practice of having health care providers’ pain management strategies for the patients’ can be categorized into three different models, the biomedical model, the non-pharmacological interventions model, and the alternative medicine model, including acupuncture[82-84],the identified factors are pivotal for advancing pain management practice level of health care providers. Training increase job satisfaction and morale that improve productivity and earn more profit which in turn advances

practicing level of health care providers’ pain management.

Thank you

**Reviewer #2: **Overall comments:

The manuscript’s strengths include the study of pain management practice and associated factors using a systematic review and meta-analysis. A specific strength was the relatively well-documented and presented data analysis performed in the "Results section." The weaknesses unfortunately concern the methodology. There are many aspects that compromise the reliability and replicability of the data. Also, the manuscript is full of grammatical errors. The manuscript requires a grammatical and language edition. Please refer to the attached comments. Thank you.

6. PLOS authors have the option to publish the peer review history of their article (what does this mean?). If published, this will include your full peer review and any attached files.

Reviewer #1: No

Reviewer #2: No

---

## [Author Response · Author response to Decision Letter 0]

5 Jul 2024

Thank you for your in-depth review and valuable feedback on our manuscript.as much as we can we tried to improve punctuation, wording, spelling, and grammar errors throughout the manuscript. Specifically based on your comment: Correct "Pain defined" to "Pain is defined as" in the Abstract, Change "Under treatment" to "Under-treatment.", Ensured consistency of "health care professionals" versus "health care providers.", Correct "Work load" to "Workload." and revised the sentence to avoid unnecessary spelling and improve clarity: "Advancing a national agenda to reduce inconsistencies in pain care for health policy, education, and practice is imperative."

---

## [Decision Letter · Decision Letter 1]

22 Jul 2024

PONE-D-24-12823R1Healthcare providers’ pain management practice and its associated factors in Ethiopia: A systematic review and meta- analysis.PLOS ONE

Dear Dr. Mekonnen,

Thank you for submitting your manuscript to PLOS ONE. After careful consideration, we feel that it has merit but does not fully meet PLOS ONE’s publication criteria as it currently stands. Therefore, we invite you to submit a revised version of the manuscript that addresses the points raised during the review process.  

We look forward to receiving your revised manuscript.

Kind regards,

Dereje Zewdu Assefa, BSc, MSc

Academic Editor

PLOS ONE

Journal Requirements:

Reviewers' comments:

Reviewer's Responses to Questions

**Comments to the Author**

1. If the authors have adequately addressed your comments raised in a previous round of review and you feel that this manuscript is now acceptable for publication, you may indicate that here to bypass the “Comments to the Author” section, enter your conflict of interest statement in the “Confidential to Editor” section, and submit your "Accept" recommendation.

Reviewer #1: All comments have been addressed

Reviewer #2: (No Response)

2. Is the manuscript technically sound, and do the data support the conclusions?

Reviewer #1: (No Response)

Reviewer #2: Partly

3. Has the statistical analysis been performed appropriately and rigorously? 

Reviewer #1: Yes

Reviewer #2: Yes

4. Have the authors made all data underlying the findings in their manuscript fully available?

Reviewer #1: Yes

Reviewer #2: Yes

5. Is the manuscript presented in an intelligible fashion and written in standard English?

Reviewer #1: Yes

Reviewer #2: No

6. Review Comments to the Author

Reviewer #1: Dear PLoS ONE team of editorials

Thank you for the chance given to me to review a manuscript entitled “Healthcare providers’ pain management practice and its associated factors in Ethiopia: A systematic review and meta- analysis “PONE-d-24-12823”. All the comments were completely addressed by the author. I recommended the acceptance of the paper .

Reviewer #2: (No Response)

7. PLOS authors have the option to publish the peer review history of their article (what does this mean?). If published, this will include your full peer review and any attached files.

Reviewer #1: No

Reviewer #2: No

---

## [Author Response · Author response to Decision Letter 1]

30 Jul 2024

Authors’ response: Thank you for your feedback regarding the CoCoPop framework. We have now incorporated the CoCoPop framework into the revised manuscript as recommended. We have added a section to the Methods part of the manuscript that explains the CoCoPop framework and how it applies to our study. By saying we conducted a systematic review and meta-analysis of observational studies. To clarify the scope and focus of our review, we applied the CoCoPop framework, which outlines the Condition, Context, and Population relevant to our study.”(See at methods sections eligibility criteria, page number 6-7).

---

## [Editor Report · Decision Letter 2]

6 Aug 2024

Healthcare providers’ pain management practice and its associated factors in Ethiopia: A systematic review and meta- analysis.

PONE-D-24-12823R2

Dear Dr. Mekonnen,

We’re pleased to inform you that your manuscript has been judged scientifically suitable for publication and will be formally accepted for publication once it meets all outstanding technical requirements.

Kind regards,

Dereje Zewdu Assefa, BSc, MSc

Academic Editor

PLOS ONE
---

## [Editor Report · Acceptance letter]

23 Aug 2024

PONE-D-24-12823R2 

PLOS ONE

Dear Dr. Mekonnen, 

I'm pleased to inform you that your manuscript has been deemed suitable for publication in PLOS ONE. Congratulations! Your manuscript is now being handed over to our production team.

Kind regards, 

on behalf of

Professor Dereje Zewdu Assefa 

Academic Editor

PLOS ONE